# Hydrogel Tissue Bioengineered Scaffolds in Bone Repair: A Review

**DOI:** 10.3390/molecules28207039

**Published:** 2023-10-12

**Authors:** Qiteng Ding, Shuai Zhang, Xinglong Liu, Yingchun Zhao, Jiali Yang, Guodong Chai, Ning Wang, Shuang Ma, Wencong Liu, Chuanbo Ding

**Affiliations:** 1College of Traditional Chinese Medicine, Jilin Agricultural University, Changchun 130118, China; ding152778@163.com (Q.D.); zhangshuai4389@163.com (S.Z.); yjl1104481279@163.com (J.Y.); mashuang20211044@163.com (S.M.); 2College of Traditional Chinese Medicine, Jilin Agriculture Science and Technology College, Jilin 132101, China; jlaulxl1221@163.com; 3Medical Science and Technology Innovation Center, Shandong First Medical University & Shandong Academy of Medical Sciences, Jinan 250117, China; zhaoyingchun26@163.com; 4College of Resources and Environment, Jilin Agricultural University, Changchun 130118, China; cgdong1008611@163.com (G.C.); wangningmyself@163.com (N.W.); 5School of Food and Pharmaceutical Engineering, Wuzhou University, Wuzhou 543002, China; 6Scientific and Technological Innovation Center of Health Products and Medical Materials with Characteristic Resources of Jilin Province, Changchun 130118, China

**Keywords:** hydrogel, tissue bioengineered scaffolds, bone repair, nanoparticles

## Abstract

Large bone defects due to trauma, infections, and tumors are difficult to heal spontaneously by the body’s repair mechanisms and have become a major hindrance to people’s daily lives and economic development. However, autologous and allogeneic bone grafts, with their lack of donors, more invasive surgery, immune rejection, and potential viral transmission, hinder the development of bone repair. Hydrogel tissue bioengineered scaffolds have gained widespread attention in the field of bone repair due to their good biocompatibility and three-dimensional network structure that facilitates cell adhesion and proliferation. In addition, loading natural products with nanoparticles and incorporating them into hydrogel tissue bioengineered scaffolds is one of the most effective strategies to promote bone repair due to the good bioactivity and limitations of natural products. Therefore, this paper presents a brief review of the application of hydrogels with different gel-forming properties, hydrogels with different matrices, and nanoparticle-loaded natural products loaded and incorporated into hydrogels for bone defect repair in recent years.

## 1. Introduction

At the present time, the difficulty in healing critical bone defects has had a great impact on the psychological health and socioeconomic development of patients [1]. The bones of the human body play an important role in people’s daily lives, such as their ability to support the body’s activities of daily living, protect organs, and balance calcium and phosphorus levels in the body, which account for approximately 15% of body weight [2]. Bone tissue continues to maintain its normal physiological structure and mineral content, with a balance between resorption by osteoclasts and bone formation by osteoblasts [3]. However, bone tissues are susceptible to injury, which causes bone defects mainly through trauma, malignant tumors, infections, and congenital diseases [4]. In addition, bone defects and osteoporosis caused by aging and disease also pose a problem in people’s lives [5]. Generally, smaller-sized bone defects are able to induce self-regeneration and repair by the organism itself, but large, critical bone defects are unable to regenerate themselves [6,7]. According to statistics, the repair of bone defects by bone grafting is performed millions of times per year worldwide, accounting for about 10% of all orthopedic surgeries [8]. Autologous or allogeneic bone grafting is currently the predominant therapeutic approach for the treatment of bone defects in clinical practice. However, the lack of donors, the more invasive nature of the surgery, immune rejection, and the limitations of potential viral transmission are all challenges that plague both patients and physicians [9]. In order to promote bone tissue repair as well as to avoid the drawbacks of autologous and allogeneic bone grafts, hydrogel tissue-engineered scaffolds have received widespread attention due to their excellent properties.

Hydrogel is a polymer material with a three-dimensional network structure that uses water as a dispersing medium [10], which has been a hot spot in medical tissue engineering research due to its good biocompatibility, improved slow release of loaded drugs, unique porous structure, permeability, and hydrophilicity, which enable it to mimic the natural extracellular matrix (ECM) and provide a suitable microenvironment for cells [11,12,13,14]. In addition, the high swelling rate of hydrogels can effectively absorb wound exudate from damaged tissues and reduce tissue infection. Higher porosity can promote the rate of gas exchange and nutrient transfer at the wound site, which is conducive to promoting the repair of damaged tissues [15,16]. The network structure of hydrogels has been reported to facilitate the proliferation and adhesion of osteoblasts, and the structure of hydrogels is very similar to the extracellular matrix of bone and cartilage, which makes hydrogels suitable for bone repair and regeneration [17]. Meanwhile, the excellent swelling properties of hydrogel can absorb tissue exudate from the damaged area of the tissue, thus reducing recurrent inflammatory infiltration and infection [18,19,20,21]. Hydrogels also have excellent hemostatic properties and tissue adhesion, which also positively modulate the repair of bone defects [22,23,24]. The use of proteins and polysaccharides as hydrogel substrates has become a hot topic in recent years. Protein-based hydrogels can form gels through various mechanisms, such as adsorption, electrostatic binding, hydrogen bonding, van der Waals forces, and covalent interactions. Compared with other gelation mechanisms, covalent interaction has the advantages of more stable gelation, better sustained release, and superior mechanical strength [25,26]. Due to the complex chemical structure of polysaccharides, they are usually modified to cross-link with other polymers to prepare hydrogel scaffolds for bone defect applications. The construction of chemical hydrogels usually utilizes the reaction of crosslinkers with derivatives of polysaccharides, which are initiated by light, electricity, or heat to form cross-linking networks through covalent bonds [27]. For example, the aldehyde groups of oxidized polysaccharides can react with the amino groups in other polymers to form hydrogel scaffolds [28], and the polysaccharide compounds esterified with methacrylate can form hydrogels by photoinitiation curing [29]. In conclusion, the formation of polysaccharide hydrogel can be achieved by azide-alkyneine cycloaddition, Diels-Alder (DA) reaction, Michael addition reaction, Schiff base, disulfide bond, borate ester, and coordination bond formation [30].

According to a previous study, Nabavi et al. [31] prepared a hydrogel with good porosity and swelling ability and loaded it with tacrolimus using gelatin and polycaprolactone. They demonstrated through in vivo experiments that their prepared hydrogel could promote the repair of cranial bone defects in rats. Chen et al. [32] prepared a four-armed benzaldehyde-capped polyethylene glycol and dodecyl-modified chitosan hybrid hydrogel inspired by the multiple healing mechanisms coordinated by organisms that promote wound repair by loading vascular endothelial growth factor (VEGF). Their study indicated that this hydrogel could promote wound repair by promoting angiogenesis, collagen deposition, macrophage polarization, and granulation tissue formation. Based on their gel-forming properties, hydrogels can be classified as hydrogels, injectable hydrogels, self-healing hydrogels, light-curing hydrogels, and temperature-sensitive hydrogels.

Bone tissue regeneration is a complex process that requires coordinated cellular behaviors such as bone immune response, precursor cell migration, osteoblast proliferation, differentiation, and bone remodeling. Multiple cells and bioactive substances are involved in activating the ponderous bone regeneration-related signaling pathways in spatial and temporal order to promote the bone repair process [33,34]. The bone repair cycle is shown in Figure 1. Osteogenesis-related genes play an important role in promoting the process of story repair, such as collagen I (CoL-I) [35], runt-related transcription factor 2 (Runx2) [36], osteopontin (OPN) [37], bone morphogenetic protein (BMP) [38], alkaline phosphatase (ALP) [39], and osteocalcin (OCN) [40]. Zhao et al. prepared a hydrogel based on peptide self-assembly, and in vivo experiments demonstrated that this hydrogel could up-regulate the expression of osteogenic factors, such as RUNX2, BMP2, OCN, and OPN, and promote osteogenic differentiation. In addition, after fracture occurrence or tooth extraction, the blood vessels in the bone rupture and bleed, and the damaged tissue cells and tissue exudate release a large number of pro-inflammatory factors, such as IL-1β, TNF-α, and IL-1α, which severely impede the differentiation of osteogenesis during the inflammatory phase and slow down the bone repair process [41,42,43]. Therefore, adding natural products with anti-inflammatory properties and promoting the expression of bone repair-related genes to hydrogels is the focus of current research.

In this paper, we review the relevant research on hydrogel tissue-engineered scaffolds in the treatment of bone defects and describe five types of hydrogels with different gel-forming mechanism types as well as the roles of hydrogels with different material matrices in promoting bone repair. In addition, we elaborated on and analyzed the application of natural products in promoting bone repair. Finally, we reviewed the application of nanoparticles in bone repair hydrogels, expecting to provide a reference for nanoparticle-loaded natural product hydrogels for promoting bone repair.

## 2. The Role of Hydrogels with Different Gel-Forming Mechanisms in Promoting Bone Repair

### 2.1. Physically Cross-Linked Hydrogels

At present, most hydrogels are chemically cross-linked hydrogels, and the excessive use of chemical cross-linking agents limits their application in fields such as biomedicine. Hence, the method of physical cross-linking has received more and more attention from researchers. Physically cross-linked hydrogels have the advantages of non-toxicity, good mechanical properties, strong cell adhesion, and slow degradation, which make them suitable for bone defect bioengineering scaffolds for bone regeneration [44,45,46,47]. Physical hydrogel refers to the hydrogel formed by the interaction of non-covalent forces between polymer chains, such as hydrogen bonding, ionic force, van der Waals interaction, polyelectrolyte complexation, stereocomplexation, and hydrophobic force [48]. To form a hydrogel, the physically crosslinked hydrogel polymer network needs to satisfy the following conditions: (a) having strong interchain interactions to form stable aggregates in the molecular network; and (b) encouraging water molecules to enter and stay in the polymer network. Hydrogels that meet these requirements can be prepared by non-covalent methods such as electrostatic interactions, hydrogen bonding, and hydrophobic interactions between polymer chains [49]. Among them, physically cross-linked hydrogels prepared on the basis of polyvinyl alcohol (PVA) have received wide attention [50]. PVA hydrogel is considered a very promising material for replacing cartilage tissue due to its biocompatibility, chemical resistance, swelling ability, and tribological behavior. While exposed to low temperatures, the water in solution freezes and the movement of the PVA molecular chains slows down, forming aggregation zones. The PVA chains form PVA microcrystals when they come into close contact with each other. When the hydrogel thaws, these microcrystals remain intact, forming a three-dimensional hydrogel network [51].

According to previous reports, Samadi et al. [52] prepared physically cross-linked tri-networked hydrogels using PVA, graphene, and agar by repeated freezing and thawing (Figure 2). Agar and graphene are the first physical cross-linking networks constructed through hydrogen bonding, and the microcrystals of PVA form the second physical cross-linking network. Meanwhile, a considerable portion of polymer chains are physically adsorbed on the surface of graphene nanosheets by forming hydrogen bonds, which is the third physical crosslinking network. The results of mechanical and self-healing tests revealed that the incorporation of PVA greatly improved the mechanical strength of the hydrogels. In addition, the tensile strength of the tri-network hydrogel was 1157 kPa, and the strain was close to 500%; moreover, the hydrogel also possessed self-healing properties; therefore, this tri-network structure hydrogel meets the requirements of tissue engineering scaffolds for bone repair, and it can be applied in bioengineering such as cartilage repair. Similarly, Schweizer et al. [53] developed a PVA-based hydrogel as a replacement for cartilage by comparing the casting-drying method with repeated freeze-thawing, and their work showed that the properties of PVA-based physically crosslinked hydrogels can be easily tailored by adapting the production method or by combining the PVA with other compounds to produce a material that most closely resembles human cartilage and that can be used as a replacement for articular cartilage tissue.

In addition, chitosan-based, physically cross-linked hydrogels are also a hot topic of current research. However, the chitosan-based, physically cross-linked hydrogels reported so far are limited in tissue engineering repair due to their low toughness and short in vivo duration [54]. In order to improve the shortcomings of chitosan-based physically cross-linked hydrogels, researchers prepared chitosan-poly(vinyl alcohol)-physically cross-linked dual-network hydrogels by repeating three cycles of freezing and thawing, and then hydroxyapatite nanocrystals superimposed on the surface layer of the hydrogel were prepared by the in-situ mineralization method, and it was found through the performance investigation that the new hydrogel had the characteristics of high strength, high porosity, and biodegradability, which could promote the repair of rabbit femur lateral condylar bone defects with potentials for use in the repair of bone tissue [55].

According to reports, collagen II is a cartilage ECM molecule found mainly in cartilage and developing bone and is thought to play an important role in both fracture healing and long bone development, and loading collagen II into tissue bioengineered scaffolds can effectively promote bone regeneration [56]. Additionally, some in vitro experiments have confirmed that collagen II can induce osteogenic differentiation of MSCs [57]. Lan et al. [58] formulated a double-crosslinked network hydrogel with PVA/collagen II as the cartilage layer and PVA/biphasic calcium phosphate/carbon nanotubes as the bone layer. The bilayer hydrogels exhibited good mechanical properties (tensile modulus up to 7.14 ± 3 MPa). In addition, they evaluated the biocompatibility of the hydrogels in vitro using two types of cells, and in vivo experiments demonstrated that the prepared hydrogels could induce the formation of cartilage regeneration.

Moreover, there are many studies on the preparation of hydrogel tissue engineering scaffolds for bone repair by physical cross-linking (Table 1). Compared with other hydrogel-forming mechanisms, physically crosslinked hydrogels have good mechanical properties and a slow degradation rate, which make them suitable for tissue engineering scaffolds for bone repair. However, physically cross-linked hydrogels are generally not suitable for irregular bone defect models and have limitations in application. Therefore, it is important to overcome the shortcomings of physically cross-linked hydrogels and develop injectable hydrogels, self-healing hydrogels, light-curing hydrogels, and temperature-sensitive hydrogels for bone repair applications.

### 2.2. Injectable Hydrogels

The development of injectable hydrogel scaffolds to effectively heal and regenerate defective bone tissue following minimally invasive implantation procedures has received considerable attention in recent years. Such scaffolds offer several advantages, as they can be injected into irregularly shaped defects and act as low-density aqueous reservoirs containing the components needed to repair and enhance bone tissue. Injectable scaffolds also promote wound healing and resultant scarring when delivered to the target site through minimally invasive surgical procedures [66]. Injectable hydrogels are in a flowable state prior to injection and can be injected through a syringe, thus having flow properties [67,68], and then the injected fluid becomes a gel in situ, resulting in the formation of tissue bioengineered scaffolds used for cell proliferation, differentiation, and adherence for the formation of new bone tissue. Such scaffolds are able to fill regular or irregular bone defects and are of great clinical importance [69]. Additionally, injectable hydrogels can also promote wound healing, tendon, and ligament repair through the delivery of natural products [70,71].

Liu et al. [72] were inspired by mussel materials to decorate nanohydroxyapatite with dopamine to form polydopamine-modified nanoparticles, and then the nanoparticles were added to sodium alginate oxide and gelatin to prepare injectable hydrogels via the Schiff base reaction. The results of in vitro experiments showed that the prepared injectable hydrogel had good bioactivity, promoted the proliferation and differentiation of bone marrow mesenchymal stem cells, and could promote the repair of rabbit bone defects in the in vivo model. Similarly, Wang et al. [73] were inspired by mussel materials to prepare nano-hydroxyapatite/poly (L-glutamic acid)-dextran injectable hydrogels for a rat cranial bone defect model by Schiff base reaction. Where the aldehyde group of aldehyde-catechol difunctionalized dextran (Dex-CHO-DP) reacts with the hydrazine group in bisphosphonyl hydrazine difunctionalized poly(L-glutamic acid) (PLGA-BP-ADH) in a Schiff-base reaction to prepare an injectable hydrogel.

Chen et al. [74] prepared magnetic hydroxyapatite/gelatin microspheres by emulsion cross-linking and incorporated them into injectable hydrogels prepared by the Schiff base reaction using carboxymethyl chitosan and oxidized gellan gum. The results of in vitro experiments demonstrated that hydrogel has excellent bacteriostatic ability, prolongs the release time of the drug, and promotes the proliferation of mouse osteoblasts, which can be used for bone repair. In addition, some researchers have also prepared injectable hydrogels by enzyme cross-linking, photocross-linking, and ultrasonic cross-linking. Zhang et al. [75] first isolated BMSC from the bone marrow of rats and used hyaluronic acid-tyramine and chondroitin-tyramine sulfate in the presence of hydrogen peroxide and horseradish peroxidase by enzyme-catalyzed cross-linking to form an injectable hydrogel tissue-engineered scaffold containing bone marrow mesenchymal stem cells (BMSCs), and the experimental results showed that this hydrogel containing BMSCs not only provided a suitable microenvironment for the adhesion, proliferation, and differentiation of the mesenchymal stem cells in vitro but also promoted bone regeneration in vivo. Ma et al. [76] used bioprinting technology to encapsulate periodontal ligament stem cells in an injectable, photocrosslinkable composite hydrogel composed of gelatin methacrylate and poly (ethylene glycol) dimethacrylate to promote alveolar bone regeneration and repair. Yuan et al. [77] developed an injectable hydrogel using silk fibroin using the ultrasonic cross-linking method. Silk fibroin hydrogel has good cytocompatibility with rabbit chondrocytes and may be a potential candidate for cartilage repair and regeneration.

In addition, many researchers have prepared injectable hydrogels for bone repair tissue engineering scaffolds (Table 2), which are clinically important and can effectively reduce patients’ pain and alleviate social pressure. However, the poor mechanical properties of injectable hydrogels and the long gelation time limit the potential of injectable hydrogels to become tissue bioengineering scaffolds; therefore, the addition of injectable hydrogels that can improve the mechanical properties as well as adjusting the appropriate ratio to promote the gelation time is the focus of the study.

### 2.3. Self-Healing Hydrogel

Hydrogels are notably characterized as bio-tissue-engineered scaffolds for tissue regeneration and drug release maintenance [86]. As a result of normal daily body movements, hydrogels may be subjected to mechanical attacks and their structure disrupted. Loss of hydrogel integrity may reduce functional efficiency and lead to loss of the hydrogel’s role as a tissue bioengineering scaffold by causing damage and the presence of cracks and cavities [87]. Self-healing materials are defined as materials that can automatically repair and restore damage [88,89]. Self-healing hydrogels allow hydrogels to self-repair within a short period of time after damage, thereby increasing the longevity and safety of the material. The self-repairing property of self-healing hydrogels improves the fatal drawbacks of poor mechanical properties of hydrogels and the inability to self-recovery after damage, which greatly facilitates the development of hydrogels into multifunctional composite hydrogels and further broadens the application of hydrogels in biomedical fields [90].

Self-healing capacity in hydrogels is chemically or compositionally doped directly into the polymer structure by doping reversible bonds (cross-linking/reacting) [91]. Self-healing hydrogels based on chitosan to repair damaged tissues have gained widespread attention [92,93]. For example, Lee et al. [94] prepared hydrogels with self-healing properties through the assembly of phytochemically modified chitosan and silica-rich inorganic nanoclay and demonstrated through in vivo experiments that the hydrogels could promote bone regeneration of non-healing cranial defects by modulating the Wnt/β-catenin signaling pathway. Li et al. [95] prepared self-healing hydrogels using PVA and methacrylate gelatin and prepared polylactide-hydroxy acetate copolymer nanofibrous membranes as a fibrous layer for bone repair scaffolds by electrostatic spinning. The mechanical and self-healing properties of the hydrogel are shown in Figure 3. These results showed that all three hydrogels were self-healing in the presence or absence of cross-linking agents due to dynamic non-covalent and covalent interactions involving the polymer-polymer network of the hydrogels. It was demonstrated by establishing a rat cranial bone defect model that this bilayer hydrogel scaffold could be used as an integrated bone grafting device with multifunctional components and has the potential to be used as a tissue-engineered scaffold for clinical bone repair. Chen et al. [96] integrated stromal cell-derived factor 1α (SDF-1α) and M2 macrophage-derived exosome (M2D-Exos) with hyaluronic acid (HA)-based hydrogel precursor solution to synthesize an HA@SDF-1α/M2D-Exos hydrogel with injectable and self-healing properties, and their results showed that HA@SDF-1α/M2D-Exos hydrogel can induce a local antimicrobial microenvironment favorable for fracture healing and has good antimicrobial activity and biocompatibility. One point to consider in their study is that the controlled release of SDF-1α accelerated the migration of BMSCs and human umbilical vein endothelial cells (HUVECs), whereas M2D-Exos improved cell proliferation, BMSC mineral deposition, and the formation of HUVEC tubes. Overall, the whole hydrogel tissue engineering scaffold was designed to complement the natural healing process of the fracture, which could limit infection while accelerating fracture healing.

In conclusion, self-healing hydrogels are suitable for potential use in bioengineered scaffolds for human tissue repair due to the specificity of their properties. However, most self-healing hydrogels are still in the basic research stage and have not been widely used in practical applications due to their poor mechanical properties and inability to adapt to the in vivo environment and the pericellular environment. Therefore, how to improve the mechanical properties of self-healing hydrogels is also one of the focuses of future research. Whether self-healing hydrogels are hazardous to humans due to the need for a chemical reaction in the hydrogel matrix still needs to be further explored.

### 2.4. Photocurable Hydrogels

Photocurable hydrogels have gained the attention of many researchers in recent years, and the photocurable reaction is usually initiated by ultraviolet (UV) or visible light. Photocurable smart hydrogels with high efficiency and strong controllability of cross-linking reactions are commonly reported in studies of tissue engineering, cell encapsulation, and drug delivery [97,98]. There are two main modes of photocuring: photopolymerization of polyfunctional monomers or mixtures of monomers and cross-linking agents into cross-linked networks, or conversion of thermoplastics into thermosets by photoreactions between polymer chains or with chains and suitable cross-linking agents [99]. Currently, photocurable hydrogels have gained widespread attention in the application of bone repair due to their ease of curing.

Zhang et al. [100] prepared a novel in situ photo-triggered-imide-crosslinked (PIC) three-component biomimetic composite hydrogel using HA, gelatin, and hydroxyapatite nanoparticles (n-HAp) as raw materials. The bionic composite hydrogel-forming mechanism involved grafting o-nitrobenzyl derivatives (NB) onto HA (HA-NB), followed by photoexcitation of o-nitrobenzaldehyde under 365 nm UV irradiation, and the subsequent reaction with gelatin-bearing amino groups to construct the hydrogel via imine linkage (HA-NB/gelatin/n-HAp). Micro-CT, fluorescent labeling, and histological observations showed significant enhancement of new bone in the composite hydrogel group, demonstrating that the photo-triggered-imine-crosslinked HA-NB/gelatin/n-HAp hydrogel can be used as a bone defect repair application.

The use of acrylamide-based polymers to modify polymers for the preparation of hydrogels has received extensive attention from researchers. Acrylic is a monomer that is cross-linked to produce hydrogels with high water-absorbent capacity as a single or multi-component system. Acrylic acid has a carboxylic acid group with the carboxyl end attached to the vinyl group. Acrylamide-based polymers can be used to prepare hydrogel tissue bioengineering scaffolds by photocross-linking, among which methacrylamide in bone repair hydrogels is of wide interest. For example, Xing et al. [101] prepared carboxymethyl chitosan methacrylate (CMCS-MA) by modifying carboxymethyl chitosan and grafting a photosensitive methacrylate group (MA) to obtain CMCS-MA, which was cured by ultraviolet light of a specific wavelength. The preparation schematic of this hydrogel is shown in Figure 4. This new CMCS-MA hydrogel has rapid light curing, good biocompatibility, bacterial inhibition, and the appropriate degradation rate, which possesses the prospect of promoting convenience and flexibility in periodontal tissue regeneration. Wu et al. [102] prepared a light-curing bilayer hydrogel scaffold for the repair of osteochondral defects in rabbits. The cartilage layer of this bilayer hydrogel is similar to natural cartilage in surface morphology and mechanical strength, and the porous subchondral bone layer loaded with human bone morphogenetic protein-2 (BMP-2) promotes the osteogenic differentiation of bone marrow stromal cells (BMSCs). Second, they developed a silk fibroin methacrylate sealer (Sil-MA) loaded with transforming growth factor β3 (TGF-β3) to promote chondrocyte migration and differentiation. Their findings suggest that the novel method of sealing Sil-MA hydrogel around the edge of the cartilage layer of the bilayer scaffold has great potential for clinical application in osteochondral regeneration. The use of methacrylate-modified polymers for the preparation of light-curing hydrogels has received much attention from researchers. For example, Wu et al. [103] injectable and light-curing hydrogel tissue engineering scaffolds based on alginate methacrylate, alginate-grafted dopamine, and polydopamine-functionalized Ti_3_C_2_ MXene (MXene@PDA) nanosheets have been reasonably designed for near-infrared-mediated bone regeneration, synergistic immune regulation, osteogenesis, and the elimination of bacteria.

In a word, photocurable hydrogel may become an excellent scaffold for bone repair, but the demand for photoinitiators limits its application in some fields, and the selection of photoinitiators becomes a key factor in determining the polymerization efficiency and the required light wavelength. Therefore, these factors must be considered before preparing photocurable hydrogels.

### 2.5. Temperature-Sensitive Hydrogels

In recent years, temperature-sensitive hydrogels have been widely used in tissue repair engineering based on their temperature sensitivity. Such as nerve repair [104], treatment of periodontitis [105], cardiac tissue repair [106], skin repair [107], and bone repair [108]. Temperature-sensitive hydrogels for cartilage tissue engineering have many advantages: (1) drugs can be easily encapsulated in the gel; (2) thermosensitive hydrogels can fill irregular cartilage defects and prevent undesirable diffusion of precursor fluids; and (3) they can easily trigger gelation under mild physiological conditions compared to other injectable hydrogels, avoiding any organic solvent damage to tissues [109,110].

Temperature-sensitive star-shaped poly-b-methoxy polyethylene glycol block copolymers (PLGA-mPEG) have good biodegradability. In one study, PLGA-mPEG block copolymer microspheres loaded with vascular endothelial growth factor (VEGF) were compounded with vascular endothelial cells to form a hydrogel, and in vivo experiments demonstrated that injectable temperature-sensitive hydrogel-loaded VEGF microspheres could be used for vascularization and bone regeneration in femoral head necrosis [111]. Sodium β-glycerophosphate (β-GP) has been shown to be one of the potential candidates for the preparation of temperature-sensitive injectable hydrogels [112,113,114,115]. In addition, to enhance the bioactivity of hydrogel scaffolds, β-GP matrix temperature-sensitive hydrogels are often used in combination with chitosan [116]. Wang et al. [117] used chitosan in combination with β-GP to prepare a temperature-sensitive hydrogel for the promotion of periodontitis repair and bone regeneration, and their research found that berberine thermosensitive hydrogel may be an effective treatment for periodontitis, which can exert anti-inflammatory and osteogenic effects through the PI3K/AKT signaling pathway. Lu et al. [118] prepared a thermosensitive hybridized hydrogel scaffold using collagen I (Col-I) and chondroitin sulfate (CS) as a matrix crosslinked with genipin, which has the advantages of being injectable, temperature-responsive, rapidly crosslinked, and biocompatible, which are favorable for clinical applications. In addition, they demonstrated in their previous report that the deletion of *Stat3* impaired the osteogenesis of mesenchymal progenitor cells in vivo and in vitro [119], so they explored the bone-repairing effect of the hydrogel by knocking out the *Stat3* gene in mice. Therefore, they investigated the bone repair effect of hydrogel by knocking out the *Stat3* gene in mice. The use of hydrogel significantly improved the healing of bone defects, as demonstrated by the experimental results (Figure 5), and the role of hydrogel in promoting bone repair was also demonstrated by the histopathological staining results and micro-CT results.

Based on previous studies showing that poloxamer is also a temperature-responsive material, poloxamer temperature-sensitive hydrogels can be used as tissue engineering scaffolds for repairing damaged tissues by transforming the solution into a gel when close to the body’s temperature [120,121]. Liu et al. [122] grafted poloxamer onto alginate and combined optimally synthesized alginate-poloxamer copolymers with filipin proteins in order to prepare temperature-sensitive hydrogels with covalent and physically cross-linked networks. They found that the formulated temperature-sensitive hydrogels could undergo a sol-gel transition at near-physiological temperatures and pH values and demonstrated in vitro results that this temperature-sensitive hydrogel could support the ability of chondrocytes to grow inward while effectively maintaining their chondrogenic phenotype. Therefore, this temperature-sensitive hydrogel has the property of becoming an alternative biomaterial for cartilage tissue engineering. In addition, osteoporosis leads to poor osseointegration and decreases implant stability. Fu et al. [123] promoted bone regeneration through the preparation of a poloxamer temperature-sensitive hydrogel loaded with simvastatin and demonstrated through in vivo experiments that the simvastatin-loaded temperature-sensitive hydrogel increased the volume fraction, thickness, and number of trabeculars, decreased trabecular segregation, and that the rate of de Osteo formation and mineral deposition was significantly increased in the treatment group.

In a word, based on the characteristics of thermosensitive hydrogel, thermosensitive hydrogel can be used as a tissue engineering scaffold to repair bone defects and treat various types of bone defects. In addition, temperature-sensitive hydrogel can be injected into the damaged, irregular human tissue to repair the damaged tissue. Compared with other synthetic polymers or modified hydrogel matrices, temperature-responsive hydrogel can avoid the harm of chemical reactions to the human body. However, the gelation rate and mechanical strength of temperature-responsive hydrogels are still important factors limiting their development.

### 2.6. Stimuli-Responsive Hydrogels

Stimulus-responsive hydrogels can promote tissue repair by controlling drug release by detecting environmental changes in the body [124,125]. Enzyme-stimulated responsive hydrogels bind enzymes directly to polymers through covalent bonding or encapsulation, and they can also interact directly with enzyme-reactive hydrogel polymers [126]. BMP, as an osteogenesis-related gene, can induce the osteogenic differentiation of mesenchymal stem cells (MSCs) to promote bone repair. It has excellent potential to promote bone defect repair when loaded into hydrogel scaffolds as an active factor. It is beneficial to repair bone defects [127].

In addition, pH-responsive hydrogels have been widely used in tissue repair engineering. pH-loud stimulus-responsive hydrogels have a controlled release mechanism in which the drug is released on demand when the pH of the body is less than the normal value of the body, thus achieving the purpose of controlled release [128]. Components with pH sensitivity are added to pH-responsive hydrogels. They include polyacrylic acid, sulfadimethoxine oligomer (SMO), polyelectrolyte N-palmitoyl CH, and oligosulfamethazine [129]. Hydroxyapatite (HAP) is an inorganic component of bone, making up 60% of natural bone. When HAP is added to pH-stimulation-responsive hydrogels, it promotes cell proliferation as well as the expression of late osteogenic markers, evidence that suggests the potential of such pH-responsive hydrogels to promote bone repair [130].

The application of ROS-responsive hydrogels in tissue repair is currently a topical area of research. When a bone defect occurs, a large amount of ROS is generated in the organism, and the excess of ROS leads to sustained cell/tissue damage and induces an amplification of the inflammatory cycle, which can further destroy the bone [131]. Hydrogel matrices can be designed to contain redox-sensitive components such as disulfide, tellurium, and diselenide bonds. These components can be broken down in the presence of reducing agents such as glutathione and dithiothreitol to control the degradation of the material and the release of drugs, growth factors, and cells [132]. In addition, the borate bond, as a covalent bond, also has ROS-responsive properties, and hydrogel coatings with borate bonds have been shown to have slow degradation in simulated ROS environments through in vitro simulated release experiments and to promote osteoblast proliferation and repair in a rat femoral defect model in vitro and in vivo [132].

Bone repair is often accompanied by inflammatory responses, and hydrogel scaffolds containing drug-loaded magnetic microspheres can inhibit bacterial growth and reduce inflammatory responses [74]. Bioactive factors can be combined with magnetic nanoparticles, which are guided and intelligently delivered to specific areas in the presence of an external magnetic field. Magnetic fields can act as external stimuli to induce specific biomechanical signals that modulate human cell behavior, such as proliferation, differentiation, or apoptosis [133]. Previous studies have demonstrated that magnetic nanomicrospheres Fe_2_O_3_ can promote the proliferation of BMSCs, and the incorporation of magnetic nanomicrospheres Fe_2_O_3_ and HAP into hydrogel scaffolds with PVA matrix can promote the expression of chondrocyte-associated osteoblastic genes, which has the potential to repair bone defects [134].

Electroresponsive hydrogels possess good response time, deformation, and memory. To date, electrically responsive hydrogels have been widely used in several smart device fields, such as sensors, membrane separation devices, and drug delivery systems [135]. The commonly used conductive materials in hydrogels are mainly conductive polymers, such as polyaniline (PAn), polypyrrole (PPy), polythiophene (PTh), polyphenylene methylene (PPv), and their copolymers and derivatives. In addition, metal nanoparticles and carbon-based nanoconductive materials such as graphene and carbon nanotubes (CNTs) have been used in hydrogel materials [136]. Conductive hydrogels have been shown to promote tissue repair [137], and the doping of magnesium-modified black phosphorus (BP@Mg) in methacrylate-modified gelatin can provide hydrogel scaffolds with photothermal conductivity. Hydrogel scaffolds in this system possess strong antimicrobial activity, improve the inflammatory microenvironment, and reduce bacterial-induced damage to bone tissue. Additionally, this photothermal-to-store hydrogel can promote the growth and migration of osteoblasts and can promote the repair of bone defect sites in an infected cranial defect model.

For complex in vivo environments, a single stimulus-responsive hydrogel is no longer able to meet the requirements of repairing tissues, and different stimulus-responsive hydrogels can be used in combination to guide the controlled release of drugs more accurately from hydrogel scaffolds. Microenvironmental disease changes favor the design of responsive hydrogels that work only in specific pathological states, and the premature degradation of hydrogel scaffolds should be avoided, resulting in the loss of cellular support for hydrogel scaffolds and the inability to promote cellular proliferation. Thus, the selection of polymers for stimuli-responsive hydrogels is also a major challenge.

## 3. Role of Different Material Matrix Hydrogels in Promoting Bone Repair

### 3.1. Hydroxyapatite

Bone is a biological hard tissue composed mainly of hierarchically assembled nano-hydroxyapatite (HAP) and organic matrix, which account for 60% of the natural bone ECM [138]. However, injuries to bones, such as critical-sized bone defects, cannot be cured by bone regeneration itself due to their complex composition and structure, which can lead to loss of self-independence, disability, or even death if not treated appropriately. The addition of HAP to hydrogels helps to promote bone regeneration in bone defect models, and HAP promotes osteogenesis mainly by increasing the expression of bone markers such as osteopontin, osteocalcin, and alkaline phosphatase (ALP) [139].

Liang et al. [140] prepared an osteomimetic osteogenic hydrogel (BOH) loaded with HAP and demonstrated by in vitro experiments that the hydrogel could promote bone mineralization, the construction of an immune microenvironment, and angiogenesis. In vivo experiments demonstrated that BOH showed excellent osteogenic effects in vivo and could promote regeneration and reconstruction of cranial defects in rats within 8 weeks. According to the previous report, the addition of the hydroxyapatite phase to chitosan-based materials showed better cell and protein adhesion, enhanced cell proliferation, and higher osteogenic gene expression [141]. Ressler et al. [130] prepared an injectable hydrogel loaded with mesenchymal stem cells using chitosan and HAP as substrates and confirmed the role of the hydrogel in promoting osteogenic differentiation by immunostaining for the osteogenesis-related genes Runx2, type I collagen, osteocalcin, and alkaline phosphatase quantification.

### 3.2. Polysaccharide Compounds

Currently, polysaccharide-based natural products have likewise received extensive attention in tissue repair hydrogel-engineered scaffolds, such as sodium alginate (SA) [142], hyaluronic acid (HA) [143], and chitosan (CS) [144].

Chen et al. [145] developed a biomimetic injectable hydrogel system based on oxidized pectin grafted with HA-adipic dihydrazide and the oligopeptide G4RGDS, and the results of the in vitro experiments demonstrated that a certain amount of G4RGDS oligopeptide doped into the HA/pectin-based hydrogel could serve as a biologically active microenvironment to support the chondrocyte phenotype and to promote cartilage formation, which is expected to be a tissue-engineered scaffold used for the regeneration of cartilaginous tissues. Liu et al. [146] prepared SA/gelatin (Gel) hydrogel scaffolds loaded with nano-bumpy clay by 3D printing. The surface microstructure, hydrophilicity, and mechanical properties were comprehensively evaluated. In addition, BMSCs were cultured in vitro with the composite hydrogel and evaluated for proliferation and osteoblast differentiation. A rabbit tibial plateau defect model was used to evaluate the osteogenic potential of the composite hydrogel in vivo. The experimental results demonstrated that the composite hydrogel loaded with nanobumpy clay exhibited good biocompatibility and effectively promoted the osteogenesis of BMSCs. Finally, histological analysis showed that the Gel/SA/nano-ATP composite hydrogel effectively promoted bone regeneration in rabbit tibial plateau defects. In addition, dextran has been reported to be a water-soluble, non-toxic, and biodegradable polysaccharide capable of forming hydrogels with various other components [147]. Ritz et al. [148] used dextran cross-linked derivatives to make hydrogels, and in order to improve the bone repair ability of the hydrogels, they also loaded SDF-1 and BMP-2 into the hydrogels. The experimental results indicated the fundamental potential of this multicomponent polysaccharide hydrogel composite as a bone regeneration biomaterial.

### 3.3. Silk Fibroin

Silk fibroin (SF) is a typical natural protein polymer with unique chemical and physical properties. SF has the distinct advantage of excellent mechanical strength to overcome the mechanical limitations encountered in other natural polymer hydrogels [149]. SF is one of the most popular biopolymers for tissue bioengineering scaffolds and holds great promise for tissue engineering applications. According to a previous study, the incorporation of SF can promote the wound repair ability of wound dressings, cell proliferation, and angiogenesis in damaged tissues [150]. These advantages of SF are also good improvements for repairing bone defects.

Photothermal effect nanoparticle hydrogel has good photothermal, antimicrobial, tumor growth inhibition, and drug release control properties under near-infrared irradiation, which is beneficial to inhibit the growth of osteosarcoma and promote the regeneration of bone tissue [151,152]. Hao et al. [153] prepared SF-based hydrogel scaffolds loaded with nanoparticles acting with photothermal effects, and the experimental results proved that the hydrogel scaffolds have great potential as bifunctional materials for photothermal treatment of tumors and bone regeneration. Furthermore, SF can be used to prepare hydrogels with other active substances for the repair of articular cartilage damage. For example, SF and chondroitin sulfate were used to prepare tissue bioscaffolds for the repair of articular cartilage defects in articulated rabbits by salt-impregnation, freeze-drying, and cross-linking methods [154]. Similarly, one study similarly prepared SF-based hydrogel scaffolds and loaded tanshinone to promote the repair of articular cartilage defects in rabbits with hydrogel scaffolds [155].

In summary, SF is a natural product with good biosafety and can improve the mechanical properties of scaffolds after incorporation into tissue-bioengineered scaffolds. Additionally, SF has good hydrophilicity and can promote cell adhesion and proliferation, which has broad application potential. However, how to improve the SF with better water solubility is still a limiting issue for the development of the SF.

## 4. Hydrogels Loaded with Natural Product Nanoparticles for Bone Repair Applications

The natural product has good antioxidant, anti-inflammatory, antibacterial, tumor inhibition, anti-osteoporosis, and tissue growth-promoting pharmacological effects [156,157,158,159,160]. Loading natural products into hydrogels with different properties can improve the bioactivity of tissue-engineered scaffolds to promote tissue regeneration and repair [161,162]. However, some natural products have some drawbacks that affect their bioavailability, such as poor water solubility, toxicity, and poor stability. Loading natural products into nanoparticles can improve the bioavailability, toxicity, and stability of drugs. Resveratrol is a natural product of trans-3,4′,5-trihydroxystilbene, which has a number of health-promoting bioactivities, but exposure to oxygen, light, temperature, and oxidizing enzymes changes the structure to cis and reduces the bioactivity of resveratrol [163]. A study encapsulating resveratrol in casein nanoparticles showed that the oral bioavailability of resveratrol when loaded into casein nanoparticles was 26.5%, which was 10 times higher than that of resveratrol when administered as an oral solution [164]. It can be seen that the addition of natural product-loaded nanoparticles to hydrogel tissue bioengineering scaffolds can effectively improve the bioavailability of natural products to promote bone regeneration.

Periodontitis is a common oral disease caused by bacteria, and its progression can lead to gum recession. Nonetheless, due to the limited regenerative capacity of periodontal bone tissue, it is difficult to promote bone tissue regeneration [165,166,167]. Therefore, tissue-bioengineered scaffolds loaded with natural products are needed to promote periodontal bone tissue regeneration. In order to avoid the side effects of conventional treatments, it has been investigated to achieve the synergistic functions of NIR photosensitization and bactericidal and periodontal tissue regeneration by gallate (EGCG) loading into gold nanoparticle-modified hydrogels (E-Au@H). In vitro research demonstrated that the NIR-irradiated composites increased the inhibition of *Escherichia coli* and *Staphylococcus aureus* biofilms by 92% and 94%, respectively, and increased the alkaline phosphatase activity of mesenchymal stem cells by 7-fold after 5 days and the mineralization rate of the extracellular matrix by 21-fold after 3 days. The results indicated that the composites could be used for the treatment of periodontal tissue regeneration with NIR-irradiated composites. The rat periodontitis model successfully demonstrated that E-Au@H irradiated with near-infrared light inhibited 87% of dental plaque and promoted alveolar bone regeneration [168].

Silymarin is a natural flavonoid lignan with excellent biological activity; nevertheless, the low water solubility of silymarin reduces its bioavailability and aqueous solubility, which limits its clinical action [169]. In order to investigate the bone-repairing effects of silymarin and to improve the bioavailability of silymarin, a study prepared silymarin-loaded chitosan nanoparticles by ionic gel technology and loaded them onto SA/Gel hydrogel tissue engineering scaffolds to evaluate their osteogenic effects. The in vitro results demonstrated that silymarin had a slow release from the scaffolds, which stimulated the differentiation of mouse mesenchymal stem cells into osteoblasts at the cellular and molecular levels [170].

In conclusion, nanoparticles loaded with natural products have been used less in hydrogel tissue bioengineering scaffolds, and nanoparticles loaded with natural products can effectively improve the slow release, bioavailability, and stability of natural products, which is clinically important for promoting bone repair.

## 5. Discussion and Future Trends

The use of hydrogels as tissue-bioengineered scaffolds for the repair of bone defects has been extensively studied, but there are still significant limitations to their clinical application. Compared with other synthetic polymer matrix hydrogels, natural product matrix hydrogels have the advantages of better biocompatibility, a simpler source, and avoiding the hazardous effects of substances produced by chemical synthesis on the human body. In addition, the purification of synthetic polymers is also a technical challenge. The different preparation methods of hydrogels with different gel-forming properties and the presence of chemical reactions limit the use of hydrogels in clinical practice, and the tissue adhesion and mechanical properties of hydrogels are also important evaluation indexes for assessing hydrogels as tissue bioengineering scaffolds for repairing bone defects. Therefore, whatever the gel-forming properties of hydrogels as scaffolds for bone defect repair, further studies are needed to address their shortcomings.

Three-dimensional (3D) and four-dimensional (4D) printing of biomaterials offers an interesting alternative for the production of allogeneic tissues and organs to circumvent the occurrence of donor scarcity and organ shortages. 3D printing allows the construction of objects by depositing materials layer by layer, allowing precise control of the dimensions and properties of complex printed structures. However, the emerging 4D printing technology allows the structure to change its shape, function, or properties over time after being exposed to specific external stimuli after fabrication, which shares some characteristics with responsive hydrogels. 4D printing of hydrogel composites is an advanced technology that can be used to fabricate scaffolds for various electrical, mechanical, and medical applications. Bioink-printed hydrogels have been extensively studied for bone repair. The hydrogel matrix can be used as a bio-glue for printing bone defects. However, due to the poor printability of hydrogels, 4D printing of detailed devices based on hydrogels remains challenging and requires improved mechanical properties and biological activity.

In addition, loading natural products into nanoparticles to solve the disadvantages of natural products and incorporating hydrogel into tissue bioengineering scaffolds for bone repair are also effective strategies to promote the bone repair ability of hydrogels. In addition, further research is needed to explore the mechanism of natural products in promoting bone repair, which can also lay the foundation for subsequent research on different types of bone defects. Hydrogels of various properties have the potential to promote bone regeneration, but they are still in the small-scale production stage in the laboratory, and the application of large-scale production is still a challenge. There are still concerns about the biocompatibility of hydrogels involving chemical reactions, so future research directions should consider the production and clinical application. This is of great significance to promote the development of green bioengineering scaffolds.

## Figures and Tables

**Figure 1 molecules-28-07039-f001:**
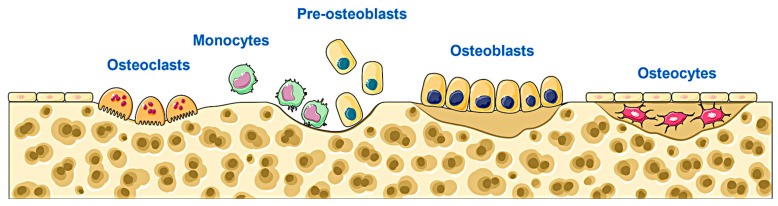
The process of osteoclasts evolving into osteocytes (the bone repair process).

**Figure 2 molecules-28-07039-f002:**
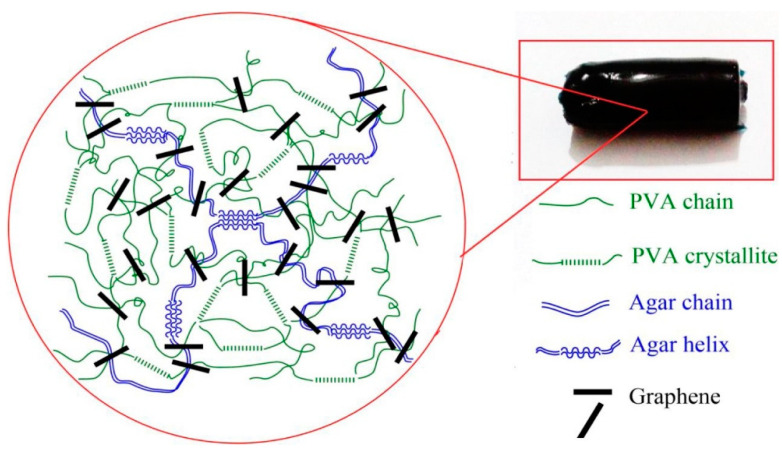
Construction of PVA hydrogel with a three-layer network structure [42], with permission from Elsevier.

**Figure 3 molecules-28-07039-f003:**
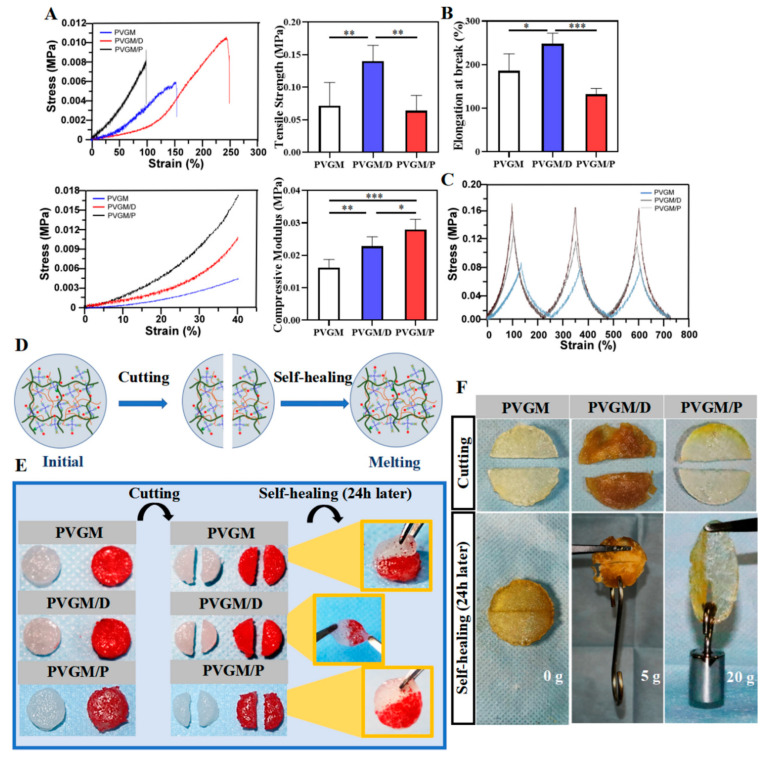
Self-healing properties of hydrogels with different crosslinkers and evaluation of compression and tension. (**A**) Compressive stress-strain curve, tensile stress-strain curve, tensile stress-strain curve, and tensile strength of the sample calculated by mechanical testing of the hydrogel. (**B**) Elongation at break (*** *p* < 0.001, ** *p* < 0.01, * *p* < 0.05). (**C**) Cyclic compressive stress-strain curves of samples. (**D**) Schematic diagram of the self-healing process. (**E**,**F**) Photographs show the self-healing behavior of two sheets of hydrogel after 24 h of cutting and the mechanical properties of hydrogels. PVGM: methacrylate gelatin/polyvinyl alcohol hydrogel. PVGM/D: gelatin methacrylate/polyvinyl alcohol/3, 4-dihydroxybenzaldehyde hydrogel; PVGM/P: gelatin methacrylate/polyvinyl alcohol/4-vinylphenylboronic acid hydrogel [95], with permission from Spring Nature.

**Figure 4 molecules-28-07039-f004:**
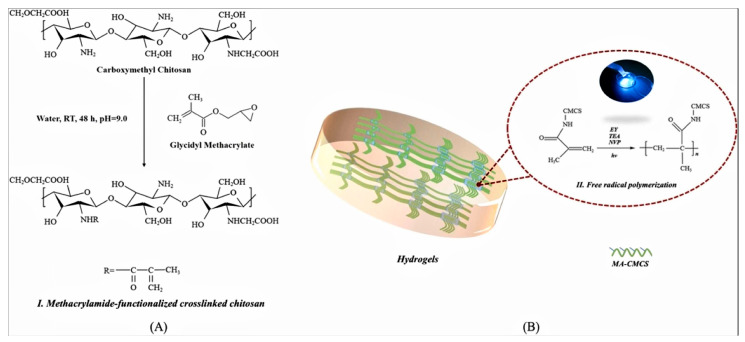
(**A**) Synthesis of carboxymethyl chitosan methacrylate (CMCS-MA). (**B**) Photocuring by photoinduced polymerization to form an in situ gel [101], with permission from Elsevier.

**Figure 5 molecules-28-07039-f005:**
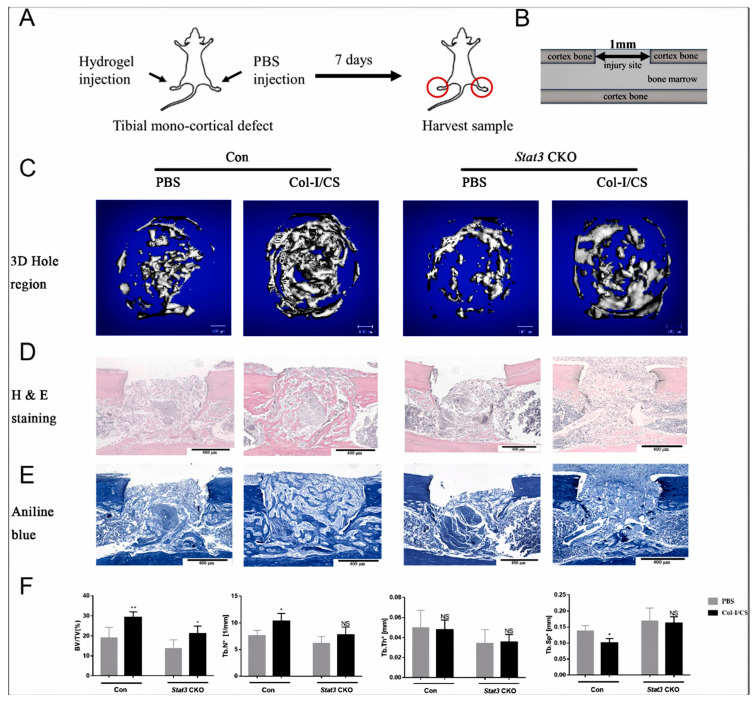
Effect of type I collagen/chondroitin sulfate hydrogel on bone repair. (**A**) Surgical flow chart. (**B**) A 1.0 mm hole was made in the tibial single cortical defect, penetrating one piece of cortical bone and entering the marrow cavity, but leaving the second cortical intact site of injury. (**C**) Micro-CT image of the bone defect 7 days after surgery. H&E staining of bone defect sections of (**D**). (**E**) Sections stained with aniline blue. (**F**) Quantitative analysis of the micro-CT index (** *p* < 0.01, * *p* < 0.05, NS: No significant difference, Comparison with PBS-treated group). BV/TV: bone volume fraction; Tb.N: number of bone trabeculae; Tb.Th: trabecular bone thickness; Tb.Sp: separation of trabecular bone [118], with permission from Elsevier.

**Table 1 molecules-28-07039-t001:** Physically cross-linked hydrogels for bone repair applications.

Hydrogel Matrix	Preparation Methods	Applications	Ref.
Nanoclay and guanidine radicalization chitosan	Self-assembly	Promoting osteogenic differentiation of MSCs	[59]
Polyetheretherketone/polyvinyl alcohol/β-tricalcium phosphate	Repeated freezing and thawing	To promote the repair of knee joint defects in rabbits	[60]
Polyvinyl alcohol/polyacrylic acid	Repeated freezing and thawing	Promoting the repair of medial condylar bone defects in rabbits	[61]
Hydroxyapatite/collagen/polyvinyl alcohol	Repeated freezing and thawing	Promoting the repair of femoral defects in goats	[62]
Methacryl gelatin/magnesium oxide	Sulfhydryl-ene click reaction	Promoting cranial bone repair in rats	[63]
Magnesium oxide/hydroxyapatite/cysteine modified γ-polyglutamic acid	The mixture was homogenized by ultrasound	Promoting tibial repair in diabetic rats	[64]
Alginate/hyaluronic acid/hydroxyapatite	Ion cross-link	It is a potential bone repair material with a good degradation rate and swelling	[65]

**Table 2 molecules-28-07039-t002:** Injectable hydrogels for bone repair applications.

Hydrogel Matrix	Methods of Preparation	Mode of Crosslinking	Applications	Ref.
Sulfhydrylated hyaluronic acid/type I collagen		Disulfide bond crosslinking	Promotes the regeneration of cartilage	[78]
Bisphosphonate modified hyaluronic acid		Non-covalent crosslinking	Promoting the repair of femoral head necrosis in rabbits	[79]
Polyethylene glycol diacrylate/sodium alginate		Photocross-link	Repair of irregular bone defects in hyperlipidemic rats	[80]
Ethylene glycol chitosan/benzaldehyde terminated polyethylene oxide derivatives		Benzoic acid-imine linkage	Promoting repair of cartilage defects in the rabbit knee	[81]
N-succinyl-chitosan/hyaluronic acid	The precursor matrix was dissolved and mixed	Schiff base reaction	Promote the survival of articular chondrocytes	[82]
Collagen/chitosan/hyaluronic acid/silica	The precursor solution was mixed	Genipin cross-linking	Promote the osteogenic differentiation of bone marrow stromal cells	[83]
Gelatin methacrylate/self-adhesive polymer	Microfluidic devices	Optical crosslinking	It has a significant therapeutic effect on the development of osteoarthritis	[84]
Gelatin-hydroxyphenylpropionic acid	The precursor matrix was dissolved and mixed	Enzyme crosslinking	To promote the repair of osteochondral defects in rabbits	[85]

## Data Availability

Not applicable.

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
