# Peer review of "Hydrogel Tissue Bioengineered Scaffolds in Bone Repair: A Review"

_molecules, 2023, doi:10.3390/molecules28207039_

Round 1

Reviewer 1 Report

This manuscript attempts to review the use of hydrogel tissue bioengineered scaffolds in bone repair. Since the most useful bone tissue engineering scaffold materials are now hydrogels (mixed with CaP, beta TCP, HA, carbonated apatite cement, or CDHA cement, etc.), this paper may be a welcome addition to the current literature in the field.

In my opinion, the idea of this manuscript is good, but unfortunately, it falls short in terms of fulfilling the essential benchmarks of a review paper.

Further, this paper contains some limitations that should be addressed. My concerns and comments are outlined below:

  1. There is no claim about what originality is compared to what has been done by others on the same matter. Are there any previous studies that have done similar work?
  2. This manuscript lacks a clear idea of the mechanism for creating hydrogels based on the polymeric nature of proteins and polysaccharides.
  3. This manuscript should include a literature review on the role of chemically cross-linked hydrogels in promoting bone repair.
  4. The manuscript lacks the applicability of other hydrogel types. The most popular section of this manuscript, "2. The role of hydrogels with different gel-forming mechanisms in promoting bone repair", needs to be more detailed regarding chemical and physical crosslinking mechanisms as well as more relevant stimuli-responsive hydrogels.
  5. In this respect, a novel cellulose-based hydrogel was developed by Weiss and co-workers that is pH-sensitive and mixed with biphasic calcium phosphate to form an injectable biphasic calcium phosphate for promoting bone repair. Considered an injectable bone substitute, it has shown good osteoconductive properties and new bone formation. For more information, please consult this reference: https://doi.org/10.1039/D2MA00410K
  6. The review will be better if there is a short comparison between protein- and polysacassaride-based hydrogels and other synthetic polymer-based hydrogels in this area.
  7. Tables 1 and 2 include a few reviews of studies. There are not enough for a review paper.
  8. The weakest component of the manuscript is the illustrative and selected figures. Thus, I suggest that the authors include more information in the figures’ captions to summarize such studies (for example, conditions, abbreviations, graphs, etc.).
  9. Since these hydrogels open doors to an exciting future where 3D/4D bioprinting could revolutionize regenerative medicine, the authors should further discuss the application of such technology by using hydrogel-based bioinks for bone repair.
  10. In the case of bone repair, do hydrogels offer a chemical environment and a surface conducive to new bone formation? This should be highlighted in a recent literature review.
  11. Since ceramics are necessary for bone repair, Section 3 (Role of different material matrix hydrogels in promoting bone repair) should include the use of other osteoconductive ceramics like biphasic calcium phosphate.
  12. A section dedicated to "Future Trends" in this area should be proposed.

Author Response

Reviewer #1: This manuscript attempts to review the use of hydrogel tissue bioengineered scaffolds in bone repair. Since the most useful bone tissue engineering scaffold materials are now hydrogels (mixed with CaP, beta TCP, HA, carbonated apatite cement, or CDHA cement, etc.), this paper may be a welcome addition to the current literature in the field.

In my opinion, the idea of this manuscript is good, but unfortunately, it falls short in terms of fulfilling the essential benchmarks of a review paper.

Response:Thanks for your comments, we have revised and responded to each of the suggestions you made.

  1. Further, this paper contains some limitations that should be addressed. My concerns and comments are outlined below:

  1. There is no claim about what originality is compared to what has been done by others on the same matter. Are there any previous studies that have done similar work?

Response:Thanks for your comments. Regarding the focus of this review is mainly on the application of natural product-loaded hydrogel scaffolds in repairing bone defect models, compared with other similar works, which usually summarise and discuss a specific type of hydrogel scaffolds used for bone repair, this review summarises the use of hydrogels with different gel properties and loaded with natural products for the repair of bone defects, which also provides a new idea for expanding the application of natural products.

  1. This manuscript lacks a clear idea of the mechanism for creating hydrogels based on the polymeric nature of proteins and polysaccharides.

Response:Thanks for your comments, it helps a lot to improve our review. We have added in the introduction section about the gelation mechanism of protein and polysaccharide matrix hydrogels.

  1. This manuscript should include a literature review on the role of chemically cross-linked hydrogels in promoting bone repair.

Response:Thank you for your comment, it provides more understanding to help us understand the types of hydrogels. Chemical crosslinked hydrogels are usually modified by chemical modification of the polymer matrix leading to the formation of hydrogels. In this review injectable hydrogels, self-healing hydrogels, light-curing hydrogels and some of the temperature-sensitive hydrogels fall within the scope of chemically cross-linked hydrogels.

  1. The manuscript lacks the applicability of other hydrogel types. The most popular section of this manuscript, "2. The role of hydrogels with different gel-forming mechanisms in promoting bone repair", needs to be more detailed regarding chemical and physical crosslinking mechanisms as well as more relevant stimuli-responsive hydrogels.

Response:Thank you for your comment, which is a great enhancement to the comprehensiveness of our review. Regarding the mechanism of chemical cross-linking we have added it in the introduction section of the manuscript. In the second part, we have added a section on the gelation mechanism of physically crosslinked hydrogels and responsive hydrogels as per your suggestion.

  1. In this respect, a novel cellulose-based hydrogel was developed by Weiss and co-workers that is pH-sensitive and mixed with biphasic calcium phosphate to form an injectable biphasic calcium phosphate for promoting bone repair. Considered an injectable bone substitute, it has shown good osteoconductive properties and new bone formation. For more information, please consult this reference: https://doi.org/10.1039/D2MA00410K

The review will be better if there is a short comparison between protein- and polysacassaride-based hydrogels and other synthetic polymer-based hydrogels in this area.

Response:Thank you for your comments, we have compared other synthetic polymer-based hydrogels, and the corresponding discussions have been added to the discussion section in this review and highlighted in yellow.

  1. Tables 1 and 2 include a few reviews of studies. There are not enough for a review paper.

Response:Thank you for your comments, we have added some studies in Tables 1 and 2 for improving the completeness of this review.

  1. The weakest component of the manuscript is the illustrative and selected figures. Thus, I suggest that the authors include more information in the figures’ captions to summarize such studies (for example, conditions, abbreviations, graphs, etc.).

Response:Thank you for your comment, we have changed the title of the chart based on your suggestion.

  1. Since these hydrogels open doors to an exciting future where 3D/4D bioprinting could revolutionize regenerative medicine, the authors should further discuss the application of such technology by using hydrogel-based bioinks for bone repair.

Response:Thanks to your comments, we have added to the discussion the prospect and discussion about the use of 3D/4D printing in bone repair. The additions are highlighted in yellow.

  1. In the case of bone repair, do hydrogels offer a chemical environment and a surface conducive to new bone formation? This should be highlighted in a recent literature review.

Response:Thank you for your comment, we have added the role of hydrogel in favour of new bone formation to the introduction, the addition is highlighted in yellow.

  1. Since ceramics are necessary for bone repair, Section 3 (Role of different material matrix hydrogels in promoting bone repair) should include the use of other osteoconductive ceramics like biphasic calcium phosphate.

Response:Thank you for your comment, the purpose of this review is mainly for future review similar to flavonoids, proteins and polysaccharides natural products in bone repair hydrogel scaffolds, for ceramics is not in the scope of this review, but it does provide a good idea for our next research-based article or review, we will seriously consider your valuable suggestions, but also for the future of hydrogel scaffolds used in bone repair research to provide a good idea for the research of the research.

  1. A section dedicated to "Future Trends" in this area should be proposed.

Response: Thank you for your comment, we have made changes and replaced "Discussion" with "Discussion and Future Trends" and the additions are highlighted in yellow in Part V.

Reviewer 2 Report

This paper presents a brief review of the application of hydrogels with different gel-forming properties, hydrogels with different matrices, and nanoparticle-loaded natural products loaded and incorporated into hydrogels for bone defect repair in recent years.Here are some suggestions to further improve the article quality.

As we have refs at the end, better to remove the author names in the Tables. change Rf to Ref.

better add a section of preparation strategies.

I dont find discussions throughout but a compilation of studies throughout. Suggest adding some discussions and comparisons.

Better write the view points in the conclusions section with the challenges and solutions for future fabrication processes.

Add most recent refs. in this field specifically in the past three years.

Author Response

This paper presents a brief review of the application of hydrogels with different gel-forming properties, hydrogels with different matrices, and nanoparticle-loaded natural products loaded and incorporated into hydrogels for bone defect repair in recent years. Here are some suggestions to further improve the article quality.

Response:Thank you for your valuable suggestions, we have made changes and replied according to your suggestions.

  1. As we have refs at the end, better to remove the author names in the Tables. change Rf to Ref.

Response:Thank you for your comment, we have made the changes you suggested.

  1. better add a section of preparation strategies.

Response:Thank you for your comment, we have added a section on preparation strategies based on your suggestion.

  1. I dont find discussions throughout but a compilation of studies throughout. Suggest adding some discussions and comparisons.

Response:Thanks for your comments, we've added a discussion and comparison in the Discussion section.

  1. Better write the view points in the conclusions section with the challenges and solutions for future fabrication processes.

Response:Thank you for your comments, we have added our views at the end of the conclusion.

  1. Add most recent refs. in this field specifically in the past three years.

Response:Thanks for your comments, we've been adding references for almost three years.

Reviewer 3 Report

1- Figure 1 has no permission.

2- Figure 1 and 2 are not adjusted in the middle of the text.

3. Preparation methods and fabrication techniques should be more explained for hydrogels.

4- Types of materials used especially natural and synthetic polymers should be added

Author Response

T1- Figure 1 has no permission.

Response:Thank you for your comment, Figure 1 was made by us without needing to get permission.

2- Figure 1 and 2 are not adjusted in the middle of the text.

Response:Thank you for your comments, we have adjusted the position of Figures 1 and 2.

  1. Preparation methods and fabrication techniques should be more explained for hydrogels.

Response:Thank you for your comment, we have added the relevant content and highlighted it in yellow.

4- Types of materials used especially natural and synthetic polymers should be added

Response:Thank you for your comments, this review focuses on the applications of natural polymers and their derivatives, for the comparison of synthetic and natural polymers has been added to the discussion section.

Reviewer 4 Report

 The manuscript entitled “Hydrogel tissue bioengineered scaffolds in bone repair: A Review” it is a well-written review, that provides the challenges associated with bone defects and the potential use of hydrogel-based scaffolds for bone repair. It also touches upon the various properties and advantages of hydrogels in this context. The review is well documented and presented in the form of figures and tables.

This work is highly suitable for publication, with minor revisions:

1. Please review the sentence in lines 58-60 as it appears to be ambiguous.

2. The quality of Figures 1 and 3 may not meet the standards for publication.

3. On line 348, the abbreviation "Col-I" is used without providing its full name.

4. The legends for Figures 1, 3, and 4 should be expanded to clarify the content and purpose of each picture (sub-figure).

5. Consider adding brief descriptions, in the text, for the figures as a single sentence may not be sufficient for the reader to understand.

6. It is advisable to italicize terms such as "via," "in vitro," "in vivo," and "et al" for proper formatting.

7. While the review provides a comprehensive overview of different types of hydrogels and their applications in bone repair, there are several areas that could benefit from further exploration or elaboration:

a) clinical translation: discuss the challenges and progress in translating hydrogel-based bone repair therapies from preclinical studies to clinical applications. Address issues related to safety, regulatory approval, and clinical trials.

b) comparative analysis: consider conducting a comparative analysis of different hydrogel types, discussing their pros and cons in the context of bone repair applications.

Minor editing of English language required

Author Response

The manuscript entitled “Hydrogel tissue bioengineered scaffolds in bone repair: A Review” it is a well-written review, that provides the challenges associated with bone defects and the potential use of hydrogel-based scaffolds for bone repair. It also touches upon the various properties and advantages of hydrogels in this context. The review is well documented and presented in the form of figures and tables.

This work is highly suitable for publication, with minor revisions:

Response:Thank you for your valuable suggestions which will be of great help in improving the quality of our review.

  1. Please review the sentence in lines 58-60 as it appears to be ambiguous.

Response:Thank you for your comment, we have double-checked the sentence and made the changes, which are highlighted in yellow.

  1. The quality of Figures 1 and 3 may not meet the standards for publication.

Response:Thank you for your comment, we have improved the quality of figure 1, but after our efforts we found that the quality of figure 3 is still poor, we have removed the information about figure 3.

  1. On line 348, the abbreviation "Col-I" is used without providing its full name.

Response:Thanks to your comment, we have added the full name of Col-I.

  1. The legends for Figures 1, 3, and 4 should be expanded to clarify the content and purpose of each picture (sub-figure).

Response:Thanks to your comments, I have expanded the legend for Figures 1, 3, and 4.

  1. Consider adding brief descriptions, in the text, for the figures as a single sentence may not be sufficient for the reader to understand.

Response:Thank you for your comment, we have added to the text accordingly.

  1. It is advisable to italicize terms such as "via," "in vitro," "in vivo," and "et al" for proper formatting.

Response:Thank you for your comment, we have changed the formatting of the relevant words.

  1. While the review provides a comprehensive overview of different types of hydrogels and their applications in bone repair, there are several areas that could benefit from further exploration or elaboration:

  1. a) clinical translation: discuss the challenges and progress in translating hydrogel-based bone repair therapies from preclinical studies to clinical applications. Address issues related to safety, regulatory approval, and clinical trials.

Response:Thanks for your comment, we've added it to the discussion section.

  1. b) comparative analysis: consider conducting a comparative analysis of different hydrogel types, discussing their pros and cons in the context of bone repair applications.

Response:Thank you for your comments, we have analysed and discussed the pros and cons of the different hydrogels in our introduction to each hydrogel, with additions highlighted in yellow.

Round 2

Reviewer 1 Report

All my comments have been addressed properly, and the corrections are acceptable.

However, concerning the properties of such hydrogels for 3D/4D bioprinting, the authors should use and add this reference to highlight this point in section “Discussion and Future Trends”.

Natural Hydrogel-Based Bio-Inks for 3D Bioprinting in Tissue Engineering: A Review. Gels 2022 (https://doi.org/10.3390/gels8030179).

Reviewer 3 Report

The authors addressed the comments properly. I recommend accepting the manuscript in its current form